# Effect of Indigenous Slaughter Methods on the Behavioural Response, Bleeding Efficiency and Cardiac Arrest of Nguni Goats

**DOI:** 10.3390/ani10020247

**Published:** 2020-02-04

**Authors:** Zwelethu Mfanafuthi Mdletshe, Munyaradzi Christopher Marufu, Michael Chimonyo

**Affiliations:** 1Animal and Poultry Science, School of Agricultural, Earth and Environmental Sciences, University of KwaZulu-Natal, P Bag X 01, Scottsville 3209, Pietermaritzburg, South Africa; zmmdletsh@outlook.com; 2Department of Veterinary Tropical Diseases, Faculty of Veterinary Sciences, University of Pretoria, Onderstepoort 0110, South Africa; chrismunya@gmail.com

**Keywords:** stress-reactions, synapse reflex, bleeding time, bleeding quality, physiological response

## Abstract

**Simple Summary:**

Resource-limited farmers under communal farming environments slaughter goats for cultural beliefs and meat consumption using indigenous slaughter methods. These methods include transverse neck incision (TNI), piercing with a short spear on the suprasternal notch targeting the heart (SNP), and piercing with a short spear under-shoulder-blade chest-floor point-of-elbow (CFP) targeting the heart to induce insensibility and death. Unsatisfied animal welfare institutes consider these slaughter methods as cruel because goats are slaughtered while sensible; therefore, experiencing pain and suffering before death. In this study, we slaughtered castrates using the above-mentioned methods and collected behavioural responses before slaughter and while bleeding, the total blood expelled during bleeding, the time it took for the blood to be expelled, the time it took the goats to lose sensibility, and time to lose heartbeat. We found that goats slaughtered using CFP lost sensibility faster. We concluded that goats slaughtered using the CFP method experienced less pain and suffering during slaughter and died faster.

**Abstract:**

Resource-limited farmers slaughter goats without stunning. The objective of the current study was to assess the influence of indigenous slaughter methods used by resource-limited households on slaughter stress-related behaviour, bleeding efficiency, and time to post-slaughter trauma of goats. Thirty clinically healthy castrated Nguni goats aged between 15 to 18 months old with body condition score of three were randomly assigned to three non-stunning informal slaughter methods, (1) transverse neck incision (TNI); (2) suprasternal notch piercing in the direction of the heart (SNP); and (3) under-shoulder-blade chest-floor point-of-elbow (CFP) sticking in the direction of the heart. Ten goats were slaughtered using each method. Slaughter method had no effect (*p* < 0.05) on stress-related behaviour. Rate of bleeding efficiency was highest (*p* < 0.05) for SNP slaughtered goats. Time to lose sensibility was lowest (*p* < 0.05) for goats slaughtered using the CFP (55 s) when compared to SNP (68 s) and TNI (75 s) slaughter methods. Time to post-slaughter trauma was highest (*p* < 0.05) for SNP (247 s) and lowest for TNI (195 s). These findings suggest that goats slaughtered with SNP experienced rapid death when compared to TNI and SNP slaughter methods. It was concluded that the SNP slaughter method is the most effective slaughter technique because it is associated with higher bleeding efficiency and lower time to lose sensibility before death.

## 1. Introduction

Slaughter is any procedure which causes the death of an animal for human consumption. Death is achieved through stunning and bleeding under conventional slaughter or only bleeding for religious slaughter [1]. In conventional slaughter methods, stunning which aims at inducing insensibility through cerebral concussion resulting in cessation of breathing and cardiac function is achieved either through a penetrative captive bolt, electrical, and gas stunning [2]. Bleeding is usually achieved through the neck incision. In developing countries, slaughter is currently one of the growing areas of public concern where cultural and religious practices are in conflict with conventional slaughter practices. The public concern is expected to further increase because animal welfare institutes enforce slaughter laws without respecting the religious needs of other ethnic groups where slaughter has a spiritual aspect.

In Southern Africa, the regions total goat population was estimated at 10,000,000 of the 422,738,294 goats in Africa [3]. Approximately 50% of the region’s total goat population is kept and owned by resource-limited households. These resource-limited households rank goats second for their socio-economic and cultural importance [4], following cattle. The socio-economic and cultural importance of goats includes selling them as live animals for an additional source of income and utilizing them for traditional ceremonies and meat consumption [5]. Though resource-limited households rear goats for selling as live animals and meat consumption, however, the primary reason for rearing goats is to perform traditional ceremonies where slaughter is done informally either using traditional [6,7] or indigenous [8] slaughter methods. In the KwaZulu-Natal province, for example, resource-limited households slaughter goats either using the transverse neck incision (TNI) or indigenous slaughter methods, which involve the use of a spear in the suprasternal notch piercing to the direction of the heart (SNP) and chest-floor point-of-elbow piercing to the direction of the heart. The selection of a slaughter method is based on time to lose sensibility and cardiac arrest.

Several studies have compared the influence of slaughter methods on the behavioural response before slaughter and while bleeding [9,10,11], bleeding efficiency [12,13], time to lose sensibility [14,15,16], and cardiac arrest [12]; however, many of these studies have been concentrated on conventional slaughter methods [9,17]. Conventional slaughter methods rely on stunning methods, which render the animal insensible before bleeding is initiated, whereas slaughter for resource-limited households is influenced by cultural beliefs, and therefore, slaughter is done without stunning either using traditional or indigenous methods. The influence of slaughter methods on the behavioural response, while bleeding, bleeding efficiency, and cardiac arrest is fairly well understood utilising conventional slaughter methods under abattoir conditions. Limited research, if any, has been conducted to assess the influence of indigenous slaughter methods on these parameters under field conditions. This research topic is very important because resource-limited farmers perceive indigenous slaughter methods as the most effective way to induce insensibility faster. It is essential to assess the influence of indigenous slaughter methods on the behavioural response and the quality of death for policy-makers to introduce policies and guidelines to improve animal welfare that are compatible with cultural requirements.

The objective of the study was, therefore, to assess the influence of indigenous slaughter methods on the behavioural response before slaughter and while bleeding, bleeding efficiency, and time to cardiac arrest for Nguni goats. It is hypothesised that the indigenous methods are just as good as the TNI slaughter referring to these relevant aspects.

## 2. Material and Methods

### 2.1. Compliance with Ethical Clearance

Ethical clearance (AREC/001/018D) for the study was granted by the University of KwaZulu-Natal.

### 2.2. Goats and Experimental Design

Thirty clinically healthy male castrates Nguni goats with a body condition score of at least 3 were used for the study. These goats were randomly bought from the local community and classified as Nguni based on their coat colour pattern and small and compact frame size. They were randomly assigned to each slaughter treatment group. Age of goats was between 15 to 18 months and was verified using dentition [18]. The body weight ranged between 18 and 21 kg (averaging 16.8 ± 1.84 kg). The body weight was measured by weighing each goat with a hanging scale (Salter Brecknell, model 235-6s, Gauteng, RSA) to the nearest 0.5 kg. Before weighing each goat, a polypropylene wooden sack bag was weighed using a hanging scale (Salter Brecknell, model 235-6s, Gauteng, RSA).

### 2.3. Treatments

A day before slaughter, all goats were detained in a kraal under the shade overnight where clean water was provided ad libitum. Goats were randomly assigned to each slaughter method and slaughtered without stunning either using a sharp knife or a short spear (refer to Figure 1A) specifically designed for the indigenous slaughtering of goats. All slaughter procedures were performed by the same skilled slaughtermen who were goat experts. The transverse neck incision (TNI) technique was performed by five slaughtermen. Four slaughtermen held a leg straight towards one side, causing the goat to lie on the floor with its back against the ground. The fifth slaughter manhandled the head and performed the neck incision using a handheld sharp knife. The knife severed the skin, muscles (brachiocephalic, sternocephalic, sternohyoid, and sternothyroid), trachea, oesophagus, carotid arteries, jugular veins and the major, superficial and deep nerves of the cervical region at the high neck cut. An average of 4 to 5 cuts was done during neck incision to each goat. Exsanguinated blood was collected using a 5-litre water bucket. After each slaughter, knives were sharpened with a knife sharpener stone (Whetstone Cutlery 20-10960 Knife Sharpening Stone-Dual Sided 400/1000 Grit Water Stone-Sharpener and Polishing Tool for Kitchen, Hunting, and Pocket Knives or Blades, Claremont, RSA) at intervals throughout the slaughter period. After skinning and dressing the carcass, carotid nerves were examined by assessing the depth of the neck incision wound. Insensibility was also inspected by checking the absence of breathing movements, threat-, withdrawal-, corneal- and eyelid reflex, as indicated by [19].

The suprasternal notch piercing (SNP) to the direction of the heart slaughter method was performed by two slaughtermen using a short sharp double-edged spear designed for slaughtering goats. During slaughter, goats were allowed to stand upright with hind limbs. While the goat was standing with hind limbs, front limbs were held by the two slaughtermen. The slaughterman that performed the piercing process held one front limb with one hand, and the second handheld a spear (refer to Figure 1A). The second front limb was held by the second slaughtermen who also held the head with one horn with the mouth facing upwards, exposing the neck and chest for effective piercing. Goats were then pierced above the brisket at a central position between the two points of shoulders to the direction of the heart (Figure 1B). After piercing, goats were allowed to lie on the ground with one side. If the goat showed any signs of physical activity such as head-turning, righting behaviour, jaw tension, and kicking, the goat was again pierced more than once, allowing it to bleed-out until it lost sensibility. During bleeding, blood was collected using a 5-litre bucket. Insensibility was also inspected by checking the absence of breathing movements, threat-, withdrawal-, corneal-, and eyelid reflex, as indicated by [19]. After skinning and dressing the carcass, the chest cavity was examined to check whether other organs were injured besides the heart.

The under-shoulder-blade chest-floor point-of-elbow piercing (CFP) to the direction of the heart slaughter method was performed by five slaughtermen. The slaughter method also involved the use of a short sharp double-edged spear, which was also used in the SNP slaughter technique. Four slaughtermen held the goat in a dorsal recumbent position. The fifth slaughterman handling the spear performed the slaughter process by piercing the goat at the heart girth next to the chest floor and point his elbow to the direction of the heart. After piercing, the goat was placed into a lateral recumbent position allowing it to bleed out into a 5-litre water bucket (Figure 1C). If the goat showed any signs of sensibility as mentioned earlier, it was pierced more than once, allowing it to bleed-out until it was rendered insensible. Loss of consciousness was also inspected by checking the absence of breathing movements, threat-, withdrawal-, corneal-, and eyelid reflex, as indicated by [19]. After skinning and dressing the carcass, other organs in the chest cavity were examined to check whether if other organs were also injured.

### 2.4. Measurements

#### 2.4.1. Stress Behaviour Response

Frequency of vocalisation, involuntary urination, and defaecation was measured to assess the influence of slaughter methods on stress behavioural response. The frequency of these events was recorded manually and using a video camera (Nikon D7100, Gauteng, RSA) by two different slaughtermen who were goat experts. Each slaughterman had a camera which recorded behavioural changes before slaughter and while bleeding. Urination frequency was measured by counting the number of times that the goat urinated during slaughter. Vocalisation frequency was measured by the number of bleats each goat made during slaughter. Involuntary defaecation was also measured by counting the number of times each goat defaecated during slaughter.

Behavioural response scores for goats before slaughter and while bleeding is shown in Table 1. These behavioural scores were recorded using a camera (Nikon D7100, Gauteng, RSA) by two slaughtermen who were goat experts. These slaughtermen were trained to take recording before the slaughtering process started. Briefly, the behavioural score for goats before slaughter was awarded as follows; a score of 1 was awarded to a goat that was calm and showed no signs of aggression or movement, a score of 2 was given to a goat that showed signs of being calm but threatened, and a score of 3 for a goat that was jumping showing aggression. During bleeding, any goat that showed a sign of calmness was given a score of 1; a score of 2 was given to a goat turning its head or moving its head, and a score of 3 was awarded to goats that were kicking with either front or hind legs or both.

#### 2.4.2. Bleeding Efficiency

The amount of blood lost during slaughter was measured from each goat by collecting blood into a 5-litre water bucket to obtain the total amount of blood expelled. Blood was then transferred from the 5-litre water bucket into a transparent glass-measuring cylinder (Borosilicate 3.3 Glass, Gauteng, RSA). The blood volume in the body cavity was determined after dissecting each goat by collecting the amount of blood found in the chest cavity, using a 1000 mL transparent glass measuring-cylinder (Borosilicate 3.3 Glass, Gauteng, RSA).

#### 2.4.3. Bleeding Time

Time of blood flow after major blood vessels were severed was recorded using a stop-watch (Accusplit Pro Survivor A601X Stopwatch, Gauteng, RSA). Bleeding time at sticking was recorded as the interval between the start of blood flow and the time the blood flow changed from a constant stream into drips. If continuous bleeding restarted after a short period of bleeding, the second endpoint was taken as the recorded bleeding time.

#### 2.4.4. Rate of Bleeding Efficiency

The rate of bleeding efficiency was determined as the volume of blood (mL) expelled in a specific time. It was calculated by dividing bleeding quality (mL) with bleeding out time (s).

#### 2.4.5. Time to Lose Sensibility

Time to lose sensibility was recorded as the interval between neck cutting or piercing of the heart to the point were signs of sensibility were absent and the animal was rendered insensible and dead. Time to lose sensibility was observed by a trained observer; therefore, all goats were observed by the same observer to minimise variation caused by involving two or more observers. The following signs were monitored: the absence of rhythmic breathing movements, threat-, withdrawal-, corneal-, and eyelid reflex (eye blink from touch), natural blinking, eye tracking to a moving object, righting reflex, and nose twitching without stimulus [19]. The eyelid reflex, the presence/absence of a blinking reaction was assessed by a gentle touch of the eyelid. Time to lose sensibility was measured using a stop-watch (Accusplit Pro Survivor A601X Stopwatch, Gauteng, RSA) by observing the time interval between piercing and the absence of the eyelid reflex, after a gentle touch of the cornea with the index finger, and the presence/absence of a blinking reaction.

#### 2.4.6. Cardiac Arrest

Immediately after the slaughter procedure was completed, time to cardiac arrest was recorded as the time (s) interval between the heartbeat after the slaughtering process until the heartbeat stopped. Heartbeats were observed using a stethoscope (Cardiology Stainless Steel JT-S747PF, KwaZulu-Natal, RSA) and recording time using a stop-watch (Accusplit Pro Survivor A601X Stopwatch, Gauteng, RSA).

#### 2.4.7. Dressing Percentage

After sticking, carcasses were hoisted and skinned using sharp knives, which were rotated depending on their level of sharpness, comfort, and balance on the hand of the slaughtermen. The dressed weight was measured by weighing each dressed carcass with a hanging scale to the nearest 0.5 kg after removing all internal organs. Dressed carcasses were weighed within one hour after slaughter. The dressed carcass comprised of the body after removing the skin, head (at the occipital—atlantal joint), forefeet (at the carpal—metacarpal joint), hind feet (at the tarsal—metatarsal joint) and the viscera. Kidneys and pelvic fat were retained in the carcass, and testes and scrotal fat were also removed. Dressing percentage was calculated by dividing dressed carcass weight with slaughter weight and multiplied by 100.

### 2.5. Statistical Analyses

All data were analysed using Statistical Analysis Software Version 9.3 (SAS 2010, Cary, NC, USA). The effects of slaughter method on behavioural responses before slaughter and while bleeding, bleeding efficiency, bleed-out time, time to lose sensibility, and time to cardiac arrest were determined using PROC GLM of SAS (2010). The following model was used:$$Y_{ij}=\mu +S_{i}+\;\varepsilon_{ij}$$
where;

$Y_{ij}$ = response variables (behavioural responses before slaughter and during bleeding; bleeding efficiency; bleeding time; the rate of bleeding efficiency; time to lose sensibility; time to cardiac arrest; and dressed percentage);

μ = population means common to all observations;

$S_{i}$ = effect of the *i^th^* slaughter method;

$\varepsilon_{ij}$ = residual error.

The correlation procedure (PROC CORR) was used to establish the Pearson correlation coefficients between the bleed-out time at sticking and time to lose sensibility; bleeding time and time to cardiac arrest; and time to lose sensibility and time to cardiac arrest.

## 3. Results

### 3.1. Behavioural Responses

The slaughter method had no influence (*p* > 0.05) on behavioural changes before slaughter and during bleeding. Across slaughter methods, the behavioural changes that were most commonly observed before slaughter were: sitting (rests on thighs and front legs)—calm but threatened (score 2), and during bleeding were: head-turning/and tail movement (wagging)—aggression or panic (score 2).

### 3.2. Bleeding Efficiency, Bleeding out Time, Rate of Bleeding Efficiency, and Total Blood Volume in the Thoracic Cavity

The effect of slaughter methods on bleeding efficiency, bleeding out time, rate of bleeding efficiency, and total blood volume in the chest cavity are shown in Table 2. Correlations between bleeding out times, total blood loss at sticking, total blood volume in the chest cavity, time to lose sensibility, and time to cardiac are shown in Table 3. The slaughter method had a significant effect on bleeding efficiency, on bleeding out time, and rate of bleeding efficiency.

The slaughter methods had a significant effect (*p* < 0.05) on the bleeding out time at sticking. The bleeding out time was lowest (*p* < 0.05) for the CFP (81.8 ± 38) slaughter treatment group when compared with the TNI (100.3 ± 38) and SNP (108.8) slaughter treatment group. There was a significant correlation (*p* < 0.05) between bleeding time and time to lose sensibility (*r* = 0.37).

The slaughter method had an effect (*p* < 0.05) on the rate of bleeding efficiency. The rate of bleeding efficiency was higher (*p* < 0.05) for the SNP (5.57 ± 0.57) slaughter method treatment group when compared with the TNI (2.51 ± 0.51) and CFP (2.61 ± 0.57) slaughter methods.

The slaughter method did not affect (*p* > 0.05) the total blood volume in the chest cavity after sticking. The total blood volume in the chest cavity after sticking was significantly negatively correlated to bleeding quality (*r* = −0.76; *p* < 0.001).

### 3.3. Time to Lose Sensibility, Time to Cardiac Arrest, and Dressing Percentage

Time to lose sensibility was significantly affected by the slaughter method (*p* < 0.05). Goats slaughtered via CFP (54.9 ± 5.4) had lower (*p* < 0.05) time to lose sensibility when compared with those slaughtered using TNI (74.5 ± 4.9) and SNP (67.9 ± 5.4) slaughter methods. Time to lose sensibility had a significant correlation (*p* < 0.05) with bleeding time (*r* = 0.37). The slaughter method influenced (*p* < 0.01) time to cardiac arrest. Time to cardiac arrest was lower (*p* < 0.05) for goats slaughtered with TNI (194.6 ± 11) when compared with the SNP (247.2 ± 12) and CFP (257.4 ± 12) slaughter treatment groups. The slaughter method had no effect (*p* > 0.05) on dressed percentage. Dressing percentage was not correlated (*p* > 0.05) with all the other behavioural or bleeding parameters. The heart was the only organ injured during slaughter.

## 4. Discussion

The hypothesis of the current study was that indigenous methods are just as good as the TNI slaughter for the stress-related behavioural response, bleeding efficiency, and time to cardiac arrest. The observed similar behavioural responses in Nguni goats might likely affirm the facts from [20] that Nguni goats are temperamental animals. Such observation could also be explained by fear caused by poor human-animal relationship considering that the goats used in the current study were kept under an extensive production system with minimal human handling. Stress shown in the form of pain or discomfort with handling could have, therefore, induced aggression or panic (score 2) influenced by the constriction of peripheral blood vessels triggered by epinephrine secretion. Such observation could also be explained by a fear pheromone induced by epinephrine or cortisol in the blood, saliva, or urine of uncleaned surfaces where other goats were previously slaughtered [21] considering that slaughtering environments were less important. The behavioural response while bleeding could best be explained by neural activity nociceptive stimuli of repetition cutting actions on the skin during neck incision and the pressure applied on the skin using a spear during the piercing process [22,23]. Repetitive cutting actions during neck incision and pressure applied on the skin during piercing stimulates nerve endings through mechanical stimuli to interpret such actions as pain. Another reason to explain aggression or panic behavioural responses during neck incision is through neck stretching and changing neck dimension during pre-slaughter handling to expose soft tissues and carotid arteries before a cut is made. To reduce the pain that stimulates aggression or a panic behavioural response while bleeding when using indigenous slaughter methods, resource-limited farmers should ensure that knives and spears used for inducing bleeding in goats are sharp and twice the width of the neck to reduce the pressure and number of cuts during neck incision and piercing process. To reduce fear before bleeding Nguni goats, resource-limited farmers should be educated about proper ways of handling goats to minimise pre-slaughter stress and slaughtering of goats in clean environments, such as concrete slabs, to prevent soiling or contamination by other goats. Moreover, this will also minimise fear behaviour induced by fear pheromones stimulated by blood, urine, or saliva dribbles from goats who were previously slaughtered in the space.

Higher total blood volume expelled during bleeding for SNP was not expected as this slaughter aimed at injuring the heart with small wound size and patency, which allows less blood to be expelled during bleeding. Observed higher total blood volume expelled at sticking for the SNP slaughter method could be associated with the severing of major blood vessels instead of the heart and aggression or panic behavioural response influenced by breathing, causing the movement of blood towards sticking wound. The observed lower total amount of blood expelled at sticking for the TNI slaughter method could best be explained by shorter knife length and position of cutting causing jugular and carotid arteries running down in the neck not to be cut during neck incision. Moreover, such an observation could also be explained by false aneurysms as a result of not cutting major arteries, causing the retraction of severed arteries, creating engorgement with surrounding connective tissue sheath to form an orifice with severed arteries to stop the opening from the wound. Observed lower total blood volume expelled at sticking for the CFP slaughter method could best be explained by the shorter length of the spear resulting in not severing the heart and small wound size created by the circumference of the spear. The shortest observed bleeding time at sticking for the CFP slaughter method could best be explained by small wound size caused by the small circumference of the spear, therefore influencing small total blood volume passing through the wound. Prolonged bleeding time could also be influenced by the formation of large blood clots through the thrombosis process [24]. The finding that the rate of bleeding efficiency was highest (*p* < 0.05) for the SNP slaughter method could best be explained by the high total blood volume expelled at sticking. To improve the rate of bleeding efficiency when slaughtering goats, resource-limited farmers should ensure that goats are handled properly before bleeding to minimise the effect of stress during bleeding. Straight long sharp knives twice the width of the neck should be used to reduce the number of cuts and increase the precision of severing major blood vessels when using the TNI slaughter method to induce insensibility and death. Resource-limited farmers who commonly use the SNP and CFP slaughter methods should be educated about the importance of severing major blood vessels that induces insensibility and death instead of severing the heart so they will improve bleeding efficiency, bleeding time, and rate of bleeding efficiency. The use of long sharp knives and spears also improves the size and patency of the wound at sticking which is important for higher total blood volume expelled at sticking, bleeding time, and rate of bleeding efficiency.

The shortest time to lose sensibility observed for the CFP slaughter method could best be explained by the spear in severing the carotid and basilar arteries during the piercing process [2], resulting in a faster rate of brain ischemia and reduced amount of oxygen and glucose available for the normal functioning of neurons in the brain, therefore inducing insensibility. The prolonged time to lose sensibility for the TNI slaughter method could best be explained by sustained blood flow to the brain through the plexus connecting the vertebral arteries with the carotid rete when an aneurysm is formed from severed carotid ends of carotid arteries [25], resulting in the delayed rate of brain ischaemia. The delayed onset of insensibility could also be explained by the inaccuracy of the slaughter method where at least one carotid artery was not cut, therefore delaying bleeding time for insensibility to occur from total blood volume expelled at sticking [14]. Prolonged time to lose sensibility for goats slaughtered using the TNI slaughter method could also be explained by the occlusion and swelling of severed cephalic ends of the carotid arteries [26], which might have lasted longer, therefore sustaining cephalic blood flow. To improve time to lose sensibility, slaughtermen who commonly use the TNI slaughter method should be well trained to ensure that major blood vessels supplying blood to the brain are severed using a straight long sharp knife twice the width of the neck when performing the neck incision. Shorter time to lose sensibility for SNP and CFP slaughter methods can be achieved by educating and training slaughtermen on how to severe major blood vessels supplying blood to the heart to induce sensibility and death.

The finding that time to cardiac arrest was shortest (*p* < 0.05) for the TNI slaughter method when compared with SNP and CFP slaughter methods suggest that the observed shortest mean time to lose heartbeat for the TNI slaughter method could best be explained by ischemia caused by reduced blood flow in the brain or the functioning of neurons [2]. Reduced blood flow in the brain resulted in a reduced availability of oxygen and glucose required to maintain the normal functioning of the neural structures in the brain stem, which is essential for the heart function or the heart muscles. Consequently, a stop in the normal functioning of the heart or heart muscles resulted in a lack of oxygen caused by an end to breathing. Prolonged time to cardiac arrest for SNP and CFP slaughter methods could have been influenced by the inaccuracy of the techniques in injuring the heart, therefore maintaining heart function and breathing.

## 5. Conclusions

Nguni goats slaughtered using SNP had a significantly higher rate of bleeding efficiency, bleeding time, and rate of bleeding efficiency. Time to loss of sensibility was significantly lowest for goats slaughtered using the CFP slaughter method. Time to cardiac arrest was lowest for goats slaughtered using the TNI slaughter method. It was concluded that SNP was the most effective slaughter method considering that the Nguni goats slaughtered using this technique died faster. The assessment of the interrelationship between the rate of bleeding efficiency, time to lose sensibility, and time to post-slaughter trauma is important in terms of pain experienced by goats during slaughter.

## Figures and Tables

**Figure 1 animals-10-00247-f001:**
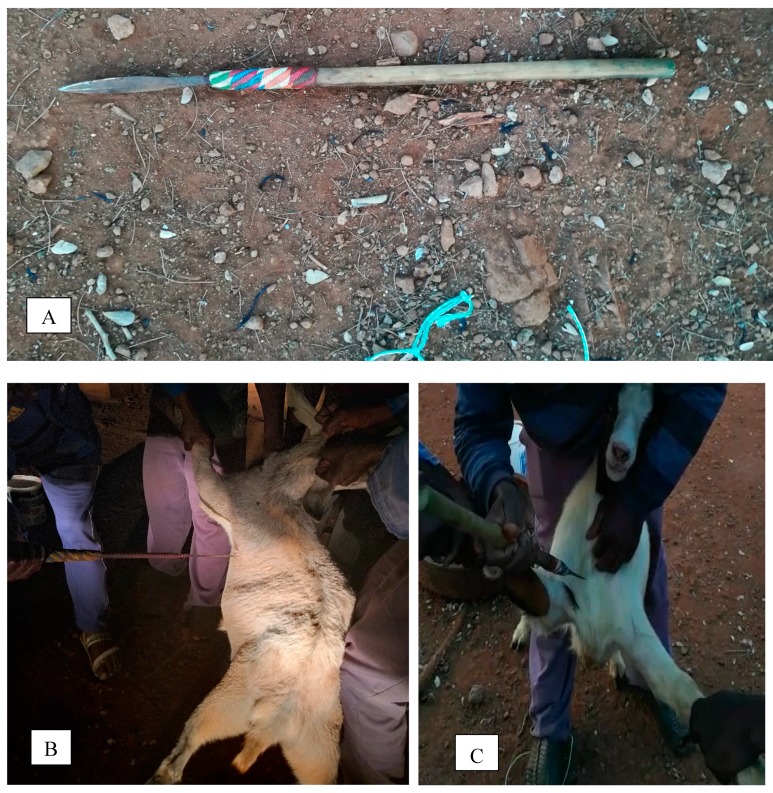
Visual pictures of a spear (**A**), suprasternal notch piercing (**B**), and chest-floor point-of-elbow piercing (**C**).

**Table 1 animals-10-00247-t001:** Stress-related behavioural scores for goats slaughtered using transverse neck incision (TNI), suprasternal notch targeting the heart (SNP), and piercing with a short spear under-shoulder-blade chest-floor point-of-elbow (CFP) methods.

Category	Score
Before slaughter	
Stable (no movement)—calm	1
Sitting (rests on thighs and front legs)—calm but threatened	2
Jumping (unstable/trying to escape/aggression)	3
While bleeding	
Stable (no movement)—calm	1
Head-turning/and tail movement (wagging)—aggression or panic	2
Kicking (front or hind legs)—aggression or panic	3

**Table 2 animals-10-00247-t002:** The effects of slaughter methods on bleeding efficiency, time to lose sensibility, and time to cardiac arrest.

Variable	Slaughter Method			Significance Level
	TNI	SNP	CFP	
Bleeding efficiency (mL)	244.6 ± 38 ^a^	566.7 ± 42 ^b^	192.2 ± 41 ^a^	**
Bleeding out time (s)	100.3 ± 7.8 ^a,b^	108.8 ± 8.7 ^a^	81.8 ± 8.7 ^b^	*
Rate of bleeding efficiency (mL/s)	2.51 ± 0.51 ^a^	5.57 ± 0.57 ^b^	2.61 ± 0.57 ^a^	**
Total blood volume in chest cavity (mL)	274.5 ± 27	ND	295.6 ± 30	NS
Time to lose sensibility (s)	74.5 ± 4.9 ^a^	67.9 ± 5.4 ^a,b^	54.9 ± 5.4 ^b,c^	*
Time to cardiac arrest (s)	194.6 ± 11 ^a^	247.2 ± 12 ^b^	257.4 ± 12 ^b^	**
Dressing percentage	51.5 ± 6.7	51.6 ± 6.7	54.2 ± 6.7	NS

Slaughter method: TNI = transverse neck incision; SNP = suprasternal notch targeting the heart; CFP = under-shoulder-blade chest-floor and point-elbow to the direction of the heart. ND: not determined. ^a,b,c^ Values in the same row with different superscripts are significantly different at *p* < 0.05; * *p* < 0.05; ** *p* < 0.01; NS—*p* > 0.05.The bleeding efficiency was lower (*p* < 0.05) for goats slaughtered using the TNI (244.6 ± 38) when compared with the SNP slaughter technique (566.7 ± 42). Bleeding efficiency did not vary (*p* > 0.05) between goats slaughtered using the TNI when compared with the CFP slaughter technique. There was a negative correlation between bleeding efficiency and total blood volume in the chest cavity after sticking (*r* = −0.76). The lower the total amount of blood expelled during exsanguination, the higher the total blood volume in the chest cavity after sticking (*p* < 0.001). Bleeding efficiency had a significant negative correlation with BTC (*r* = −0.76; *p* < 0.01).

**Table 3 animals-10-00247-t003:** Correlations between bleeding out times, total blood loss at sticking, total blood volume in the chest cavity, time to lose sensibility, and time to cardiac arrest.

Variable	BOT	TVS	BTC	TIP	TCA
BOT		0.23	−0.24	0.37 *	−0.03
TVS			−0.76 ***	−0.07	0.16
BTC				0.02	−0.08
TIP					−0.21

Significantly correlated at * *p* < 0.05, ** *p* < 0.01, *** *p* < 0.001 BOT: bleeding out time at sticking; TVS: bleeding efficiency; BTC: total blood volume in chest cavity after sticking; TIP: time to lose sensibility; TCA: time to cardiac arrest.

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
