# Peer review of "Effect of Indigenous Slaughter Methods on the Behavioural Response, Bleeding Efficiency and Cardiac Arrest of Nguni Goats"

_animals, 2020, doi:10.3390/ani10020247_

Round 1

Reviewer 1 Report

The manuscript titled “Effect of indigenous slaughter methods on the 2 behavioral response, bleeding efficiency and time to 3 post-slaughter trauma of Nguni goats” has been well articulated from the abstract to Conclusion. Materials and Methods are clear enough to allow repeatability of the research. Results were interpreted and discussed at length. I commend authors for the recent and relevant literature cited in this study.

The study is not only beneficial to scholars but to farming communities, to have a clear understating of the animal welfare issues. Furthermore, it cannot be avoided the fact that tradition will never come to an end rather we can learn from the indigenous practices and improve on them but bringing novelty out of them. This will also assist in the future to formulate datasheets to quantify the number of animals slaughtered during different ceremonies to make informed decisions towards livestock improvement, animal welfare, and production.

I, therefore, recommend the paper to be considered for acceptance in Animals after addressing minor comments summarised below.

Simple summary

Reduce lines 12-16 to produce 2 sentences. End the first sentence whit “using indigenous methods.” The second sentence should begin with, these methods include transverse……..”.

Line 17-19; rephrase to “Unsatisfied animal welfare institutes consider these slaughter methods as cruel because goats are slaughtered while sensible, therefore, experiencing pain and suffering before death”.

Line 20; Is bleeding a reflection of distress? Please rephrase this statement. Authors need to also note that animals will surely show some movements when slaughtered, however, these are limited, hence there is a need to specify “behavioral response”.

Introduction

Line 51; Change ‘were’ to ‘where slaughter has spiritual meaning’.

Author Response

Point 1: Line 12-16 has been rephrased and summarized to produce two sentences.

Point 2: Line 17-19 has been rephrased to “Unsatisfied animal welfare… suffering before death”. In line 17, “consider” was also added between “institutes” and “these”.

Point 3: The statement as a reflection of distress in line 19 has been deleted.

Point 4: In line 51, “were” was changed to “where” and added between “groups” and “slaughter”.

Reviewer 2 Report

Thank you for this very interesting paper! There is one main (global) comment I have to make: in the simple summary as well as in the introduction you are talking about public concerns and conflicts between cultural and religious practices during slaughter and conventional slaughter practices. Therefore it would be of great interest how your results relate to similar results with conventional slaughter methods. Please add results about stress before and during slaughter and about bleeding quality from other research papers to your discussion part.  There should be at least some comparison to conventional methods. Otherwise you don't have to mention them before.

Please use the words indigenous and traditional always in the same way. I don't understand why you say on line 29 that you are including three indigenous slaughter methods which are TNI, SNP and CFP, but on line 61 you say that they use Either the TNI or ones of the two indigenous methods SNP and CFP. On line 82 you say that your hypothesis is that the indigenous methods are just as good as TNI. But is TNI not an indigenous method? Then it should not be mentioneds as one of three in the abstract! Please explain it better: what is indigenous? And what is the difference to traditional? In table 1 you call them "informal slaughter methods" is informal the same as indigenous? Please explain that better or use alwaiys the same expression.

Line 17: please write: „Unsatisfied animal welfare institutes consider these slaughter methods as cruel...“

Line 27: Please write „...time to post-slaughter trauma of goats.

Lines 37/38: „It was concluded that the CFP slaughter method is the most efecctive technique (not: techniques!), because it is associated ... “

Line 46: add: „through cerebral concussion...“

Line 51: „...groups where slaughter has a spritual aspect

Libe 70.“slaughter is done without stunning ...“

Lines 74-76: please replace the whole sentence starting with „it is important“ by: This research topic is very important.“

Line 79: move „to improve animal welfare“ after the word „guidelines“.

Line 81: please add „behavioural response before slaughter“

Line 83: Please replace the last line of the introduction part by: „TNI slaughter referring to these relevant aspects“.

Line 94: please name this chapter 2.2 just „Goats“

Line 95: remove the sentence in brackets (you say the samew 4 lines further down)

Line 99: remove „for cthese goats“ and replace „18 to 21 kg“ by „18 and 21 kg“

Line 101: and a propylene wooden sack

Line 101-104: remove this sentence, it is not necessary, because it is obvious.

Line 112: introduce a space between „man“ and „handled“

Lines 124-127: please make 2 or 3 sentences out of one. I don’t understand who holds the head of the goat and how. Please describe it more precisely and not in one sentence.

Line 129: replace „with“ by „on“

Linbes 132-133: was this not done after slaughtering with the TNI-method? If yes, please describe it also there (on line 119)

Line 134: replace „assess“ by „check whether“ (the results of this check are not mentioned in the results part! This has to be done)

Line 141: point his elbow

Line 146: replace „assess“ by „check whether“ (the results of this check are not mentioned in the results part! This has to be done)

Line 150: replace the mentining of all events again by: „the fequency of these events was...“

Line 151: was it really observed by the same slaughtermen who slaughtered? How many men were there for observation? And were the frequencies counted from the video or direct? Please describe this closer!

Line 156: replace „is shown2 by „are shown“

Line 157: were there 1 or 2 camreas? (because there were two slaughter men)

Line 159: Briefly, the behavioural score...“

Line 161: replace „showing“ by „showed“

Line 170: please define: „Bleeding quality is the amount of blood expelled...“ (because it is not clear whether the blood in the body cavity is also measured for bleeding quality.

Line 190. Monitored: (remove which include) the absence...

Line 193. Please move the sentence „and the presence / absence of a blinking reaction“ to the beginning of the line, after thew word „reflex“

Line 195: by observing „the time interval between pearcing and the absence of the eyelid reflex, after a gentle touch...

Line 200: Which intervall was it: from the heratbeat after the slaughter process until heartbeat changed or until heartbeat stopped?

Line 232: How frequent were those events? Please show them in a table!

Line 236: „bleeding out time“

Line 238: „On bleeding out time“

Line 239: total amount of blood expelled (= bleeding quality)

Line 264: Please add results on injuries of other organs here!

Line 277: were those repetitions counted?

Table 2: „bleeding-out time“ Please mentkion all parameters in the title or summarize them

Line 317: „for the SNP...“

Line 318: „best be explained by the high...“

Line 325: replace „thus nimproving“ by „so they will improve“

Line 329: „The shortest time to lose sensiblity observed for the...“

Line 339: „using the TNI slaughter method...“

Line 348: suggests that the

Lines 363-364: I don’t understand this sentence.

Author Response

Point 1: informal slaughter methods has been changed and replaced with “traditional or indigenous slaughter” or “TNI, SNP and CFP” throughout the manuscript.

Point 2: “consider” was also added between “institutes” and “these”.

Point 3: post-slaughter trauma has been deleted and replaced with cardiac arrest.

Point 4: “the” has been added between “that” and “CFP”. “techniques” has also been changed to “technique” between “effective” and “this”. “because it is” has also been added between “technique” and “with”.

Point 5: “through cerebral concussion” has been added between “insensibility” and “resulting”.

Point 6: “were” was changed to “where” and added between “groups” and “slaughter”. “has a” has also been added between “”slaughter” and spiritual”. “aspect” has also been added between “spiritual” and “In”.

Point 7: “is done” has been added between “slaughter” and “without”.

Point 8: the whole sentence has been replaced with “it is important… is very important “.

Point 9: “to improve animal welfare “ has been moved after the word “guideline”.

Point 10: “behavioural response before slaughter” has been added between “on” and “and”.

Point 11: the last line has been added “TNI slaughter referring to these relevant aspects”.

Point 12: “and experimental design” has been deleted.

Point 13: the sentence in brackets has been removed.

Point 14: “for the goats… 18 to 21 kg” has been removed and replaced with by “18 and 21 kg”.

Point 15: Sentence in lines 101-4 have been removed.

Point 16: a space has been introduced between “man” and “handled”.

Point 17: Sentence in lines 142-7, have been rephrased to produce 3 sentences which clearly explains the whole slaughter process for the SNP treatment group.

Point 18: “with” has been replaced with “on” between “ground” and “one”.

Point 19: Insensibility was also… as indicated by [26].” Was also added at the end of the sentence.

Point 20: “assess” was deleted and replaced with “by check whether” between “to” and Point 21: “other”. Results for organs injured during the slaughter process has been reported in line 258/259.

Point 22: section 2.4.1 has been changed from “behavioural changes” to “stress behavioural responses”.

Point 23: “his” has been added between “point” and “to”.

Point 24: Recording of behavioral responses are clearly explained in line 142-4.

Point 25: “assess” was deleted and replaced with “by check whether” between “to” and “if”.

Point 26: the mentioning of all events has been deleted and replaced with the “the frequency of these events”.

Point 27: “shown2” has been replaced with “shown”.

Point 28: “the” was added between “briefly” and “behavioural”.

Point 29: “bleeding quality” has been rephrased to “bleeding efficiency”. Such has been done throughout the manuscript for consistency.

Point 30: bleeding efficiency has been clarified that it’s the total amount of blood expelled during bleeding. Total amount of blood collected in the thoracic cavity after dressing the carcass is not part of bleeding efficiency.

Point 31: “monitored:” had been added between “were” and “which” and “remove which include” was deleted.

Point 32: “by observing the.. a blinking reaction” has been changed to “the time interval… a gentle touch”.

Point 33: was rephrased to show that time to cardiac arrest is an interval from the heartbeat after slaughter process until heartbeat stopped.

Point 34: The frequencies of events stated in line 323 are not presented in the result section since they have no significant difference between the slaughter treatments.

Point 35: “bled-out time” has been changed to “bleeding out time”. Such change has been done throughout the manuscript for consistency.

Point 36: Results on injuries for other organs are reported in line 256-7.

Point 37: Average number of cuts for TNI slaughter method are reported in line 105.

Point 38: all parameters were mentioned in table 2. “Bled-out time” was changed to “bleeding out time” in table 2.

Point 39: “the” has been added between “for” and “SNP”.

Point 40: “by the” has been added between “explained” and “high”.

Point 41: “thus improving” was rephrased and replaced with “by, so they will improve”.

Point 42: “observed” was added between “sensibility” and “for”.

Point 43: “the” was added between “using” and “TNI”.

Point 44: “suggests” has been added between “methods” and “that”.